# Barriers from Multiple Perspectives Towards Physical Activity, Sedentary Behaviour, Physical Activity and Dietary Habits When Living in Low Socio-Economic Areas in Europe. The Feel4Diabetes Study

**DOI:** 10.3390/ijerph15122840

**Published:** 2018-12-13

**Authors:** Vicky Van Stappen, Julie Latomme, Greet Cardon, Ilse De Bourdeaudhuij, Mina Lateva, Nevena Chakarova, Jemina Kivelä, Jaana Lindström, Odysseas Androutsos, Esther González-Gil, Pilar De Miguel-Etayo, Anna Nánási, László R. Kolozsvári, Yannis Manios, Marieke De Craemer

**Affiliations:** 1Department of Movement and Sports Sciences, Ghent University, 9000 Ghent, Belgium; greet.cardon@ugent.be (G.C.); Ilse.DeBourdeaudhuij@UGent.be (I.D.B.); Marieke.DeCraemer@UGent.be (M.D.C.); 2Clinic of Paediatric Endocrinology, Medical University Varna, 9002 Varna, Bulgaria; mina_pl@yahoo.com; 3Clinical Center of Endocrinology, Medical University of Sofia, 1431 Sofia, Bulgaria; veni_chakarova@abv.bg; 4National Institute for Health and Welfare, 00271 Helsinki, Finland; jemina.kivela@thl.fi (J.K.); jaana.lindstrom@thl.fi (J.L.); 5Department of Nutrition and Dietetics, School of Health Science & Education, Harokopio University, 176 76 Athens, Greece; oandrou@hua.gr (O.A.); manios.feel4diabetes@hua.gr (Y.M.); 6Growth, Exercise, Nutrition and Development (GENUD), University of Zaragoza, 50009 Zaragoza, Spain; esthergg@unizar.es (E.G.-G.); pilardm@unizar.es (P.D.M.-E.); 7Debreceni Egyetem (UoD), University of Debrecen, 4002 Debrecen, Hungary; nanasi.anna@sph.unideb.hu (A.N.); kolozsvari.laszlo@sph.unideb.hu (L.R.K.)

**Keywords:** physical activity, sedentary behaviour, dietary habits, type 2 diabetes prevention, low socio-economic status, qualitative study, European study, Feel4Diabetes-study

## Abstract

This study investigated barriers towards health behaviours (physical activity, limiting sedentary behaviour and healthy dietary habits) experienced by young European families living in vulnerable areas, from multiple perspectives (parents, teachers, local community workers). Focus groups were conducted in six European countries (Belgium, Bulgaria, Finland, Hungary, Greece and Spain). In each country, three focus groups were conducted with parents, one with teachers and one with local community workers. Data were analysed using a deductive framework approach with a manifest content analysis using the software NVivo. The present study identified barriers on four levels (individual, interpersonal, organisational and macro level) of a socio-ecological model of health behaviour. From parents’ perspectives, both general barriers (e.g., financial limitations and lack of time) and country-specific barriers (e.g., organisational difficulties and inappropriate work environment) were identified. Additional barriers (e.g., lack of parental knowledge and lack of parental skills) were provided by other stakeholders (i.e., teachers and local community workers). The results of this study demonstrate the additional value of including multiple perspectives when developing a lifestyle intervention aiming to prevent type 2 diabetes in vulnerable groups. Future lifestyle interventions are recommended to include multiple components (family, school, and community) and could be implemented across European countries if country-specific adaptations are allowed.

## 1. Introduction

In Europe, the prevalence of type 2 diabetes (T2D) has increased dramatically. In 2017, 58 million European adults (20–79 years old) suffered from T2D and this number is predicted to be 67 million by 2045 [1]. Furthermore, the prevalence of T2D is significantly higher in areas with a low socio-economic status (SES), so-called “vulnerable areas” (e.g., areas part of a region, district or town with a high unemployment rate or a high percentage of residents with a low educational level) [2]. T2D can be linked with several negative individual health consequences (e.g., cardiovascular disease, kidney disease, and pregnancy complications) and has a considerable impact on costs for society [1]. Apart from biological factors and genetic predisposition, T2D is determined by an unhealthy lifestyle [3]. More specifically, physical inactivity, high levels of sedentary behaviour (SB) and unhealthy dietary habits independently contribute to the development of T2D [4,5,6]. In the past, SB and physical inactivity were often considered as synonyms [7]. Nevertheless, engaging in sufficient physical activity (PA) does not counterbalance the negative health consequences of excessive SB. Therefore, physical inactivity and SB should be treated as two distinct behaviours affecting health in a different way [7,8]. As unhealthy lifestyle behaviours are more prevalent in vulnerable areas [9,10], effective interventions focusing on this subgroup of the population are needed to halt the increase of T2D.

Early childhood presents a critical period to start preventive efforts, because T2D risk factors such as obesity or low levels of PA track from childhood to adulthood [11,12]. Furthermore, evidence has shown that school- and community-based lifestyle interventions starting from a young age (6–12 years) are more likely to be successful and cost-effective in reducing the long-term risk for T2D [13,14], and family-based lifestyle interventions appear to be effective in improving health behaviours of both children and adults [15,16,17,18,19]. Taken together, multicomponent (including family, school and community) lifestyle interventions aiming to prevent T2D in young families with primary school-aged children living in vulnerable areas is a public health interest. However, intervention studies targeting these families are underrepresented in the literature, thus more efforts are needed in developing interventions for these families.

A first important step in developing an intervention for this target group is to gain insight into the barriers that they experience towards healthy lifestyle behaviours. Studies investigating barriers perceived by families living in vulnerable areas from parents’ perspective show that, for example, financial constraints [20,21], parents’ busy work schedules [20], lacking time to purchase and prepare healthy foods [20,21,22], children’s resistance in consuming healthy foods [20,21], lack of support from friends and/or family in adopting healthy dietary habits [22,23] and unhealthy food preferences of both the partner and children [22] are barriers to healthy dietary habits. Lack of time [23], feelings of exhaustion [23], lack of opportunities for participation in physical activity (PA) [24], difficulties with transportation [24], lack of facilities and space at home [25], sports activities being too expensive [24] and children disliking PA [24] are barriers to sufficient PA. Barriers to limiting SB are low self-efficacy of parents in monitoring the child’s screen time behaviours and importance of TV viewing as family-time [25]. However, these studies only consider parents’ perspectives. Besides the parents, primary school teachers are well placed to comment upon potential barriers experienced by young families, because children spend many hours of the day in school, without their parents [20,26] and parents and teachers are often in contact with each other [27]. Furthermore, teachers might also be suitable to comment on barriers related to the school environment [28]. Second, families living in vulnerable areas have a close connection with the local community and often use community-based primary health care services. As local community workers (also known as community health advisors or outreach workers) provide informal community-based health-related services and establish vital links between health providers and the community [29], we suggest that also the perception of local community workers on the barriers experienced by families living in vulnerable areas might be of added value. Additionally, engaging teachers and local community workers in developing lifestyle intervention for preventing T2D might be of added value not only for their perspective on the barriers but also because both teachers and local community workers can support the development of an intervention preventing T2D in young families living in vulnerable areas.

To the best of our knowledge, no studies have been conducted investigating the barriers towards health behaviours experienced by young European families living in vulnerable areas, from multiple perspectives (i.e., parents, teachers and local community workers) and across multiple European countries representing different socio-economic levels (e.g., low-to-middle income countries vs. high income countries vs. high-income countries under austerity measures). As families living in different European countries might experience other barriers towards health behaviours because of differences in cultures, values and socio-economic levels of the countries, it is of interest to gain insight into the similarities and differences in perceived barriers between European countries. Furthermore, health and health behaviours are being affected by multiple levels of influence, including personal (e.g., psychological), environmental (e.g., both social and physical), and policy levels [30]. Understanding and studying families’ perceived barriers towards health behaviours from a broad, multilevel perspective is therefore recommended. The present study was guided by the socio-ecological model of health behaviour to investigate these barriers [31]. This model is often used in health research and offers a broad perspective on health behaviours, integrating multiple, hierarchically-nested levels of influence (i.e., the individual, interpersonal, community/organisational and public policy level) [32].

The aim of the present study was twofold. The first aim was to investigate, from a socio-ecological perspective, the barriers experienced by young families living in vulnerable areas towards healthy lifestyle behaviours according to the perspective of parents, and to explore whether these barriers are general or country-specific. The second aim was to investigate if teachers and local community workers report barriers (that they believe families are experiencing) in addition to those provided by parents, which might be of added value to guide the development of future interventions.

## 2. Materials and Methods

### 2.1. Study Background

This study was conducted as part of the European Feel4Diabetes-study, which aims to develop an evidence-based, and potentially cost-effective and scalable intervention to prevent T2D in young families living in vulnerable areas across six European countries [33]. These countries represent three socio-economic levels derived from the World Bank’s Gross National Income (GNI) index [34] and the Eurostat’s Government Budget Deficit data in 2014 [35]: high-income countries (Belgium and Finland), low-to-middle-income countries (Bulgaria and Hungary) and high-income countries under austerity measures (Greece and Spain). As the prevalence of T2D is significantly higher in (any area of) low- to-middle-income countries [2], any area in Bulgaria and Hungary was defined as a vulnerable area. In high-income countries and high-income countries under austerity measures, the prevalence of T2D is higher in low SES areas [2], and therefore only low SES areas were defined as vulnerable areas in Belgium, Finland, Greece and Spain. These low SES areas were determined based on socioeconomic indices retrieved from official sources and authorities (e.g., high unemployment rate). The Feel4Diabetes-study aims to target three main energy balance-related behaviours, i.e., PA, SB and dietary habits. To get more insight into the barriers that young families living in vulnerable areas experience towards these three behaviours, focus group interviews were conducted within each country. Within focus groups, group interactions were stimulated resulting in more information than individual interviews would provide [36]. Furthermore, focus groups are interactive compared to group interviews in which opinions from individuals within a group setting were gathered. Within focus groups, the group opinion is at least as important as the individual opinion and the group itself may take on a life of its own, not initiated by the researcher [37]. In total, five focus groups were conducted: three focus groups with parents, one with teachers and one with local community workers. All focus groups were conducted between May and June 2015.

### 2.2. Participants

For the focus groups, parents of primary schoolchildren (6–12 years old) living in vulnerable areas, and primary school teachers and local community workers who are often in contact with young families living in vulnerable areas, were recruited through schools, local community centres and medical centres located in vulnerable areas. Mostly, within focus groups, participants were connected through their recruitment channel. Between focus groups, participants were in some cases linked by being recruited in the same school, local community/medical centre or area. In each country, focus group participants (parents, teachers, and local community workers) were recruited, for convenience reasons, in one or more vulnerable areas close to the research centre (±40 km), also aiming for gender-mixed groups. More detailed information on recruitment area(s) can be found in Table 1 and Table 2.

### 2.3. Procedure

In each country, five focus groups were conducted. A protocol with three semi-structured interview guides (one for parents, one for teachers and one for community workers) was developed to ensure standardisation across the six European countries. The interview guides included three main parts: Part I, questions on families’ PA; Part II, questions on families’ SB; and Part III, questions on families’ dietary habits. Each part included questions about barriers at each level of the socio-ecological model of health behaviour (i.e., individual/interpersonal level, community/organisational level and macro/public policy level) [32]. The semi-structured interview guides for parents, teachers and local community workers can be found in Appendix A. The interview guide was pilot tested in a critical reference group of eight participants for relevance, comprehensibility and timing of the focus group discussions [38]. Before each focus group, a written informed consent form was signed by each participant in which participants’ anonymity was ensured and participations gave permission to audiotape the focus group discussion. In all countries, refreshments (e.g., coffee, juices, and water) and healthy snacks (e.g., fruit) were provided during the focus groups. Only in Belgium, Finland and Bulgaria, incentives (i.e., fruit baskets, coupons or badminton sets) were provided to increase the response rate. Across the countries, 115 parents, 45 teachers and 41 local community workers participated in the focus groups, which lasted approximately 90 min and were led by a trained moderator. A co-moderator was available to handle logistics and to assist the moderator by pointing out questions that were missed. Afterwards, a demographic questionnaire form including questions such as gender, age and education level was filled out by each participant.

### 2.4. Data Analysis

Barriers towards each of the health behaviours were identified using a deductive approach with a qualitative content analysis [39]. All conducted focus group interviews were audiotaped and each country made a written transcription of the focus group interviews in English. Then, a manifest content analysis [40] was applied to these interview transcripts using NVivo 11.0 (QSR International, Melbourne, Australia). This was done centrally to ensure reliability and comparability across countries in barrier identification. For this, one framework (coding tree) was created, including the three major topics of the semi-structured interview guide (PA, SB and dietary habits) and the four levels of the socio-ecological model. After pre-testing and modifying this coding tree based on apparent difficulties and/or divergent interpretations in its use [41], barriers were coded by two independent doctoral researchers (V.V.S. and J.L.; ICC: 0.86) to increase the comprehensivity and provide sound interpretation of the data [41,42]. This coding of barriers was then discussed between the researchers until full consensus was reached. After coding all barriers, barriers were classified as “general barriers” when they were reported by parents in four of the six countries or more, as “country-specific barriers” when they were reported by parents in only one or two of the countries, and as “additional barriers” when they were uniquely reported by teachers and/or community workers, but not by parents.

### 2.5. Ethics Approval and Consent to Participate

All applicable institutional regulations pertaining to the ethical use of human volunteers were followed during this research. Ethical approval was provided by the Ethical Committees of all participating European countries (i.e., Ethical committee of Ghent University Hospital (Belgium), Committee for the Ethics of the Scientific Studies (KENI) at the Medical University of Varna and the Municipality of Sofia (Bulgaria), Ethics Committee of Harokopio University of Athens, the Greek Ministry of Education, Research and Religious Affairs and the Municipalities of Kallithea, Peristeri, Piraeus and Keratsini-Drapetsona (Greece), CEICA (Comité Etico de Investigacion Clinica de Aragon (Spain), Ethics Committee of THL (Finland) and the Bioethics Committee of University of Debrecen (Hungary)). Participants received an information letter in which they were briefly informed about the purpose of the study and signed a written informed consent.

## 3. Results

### 3.1. Descriptives

Descriptive information of the focus groups with parents can be found in Table 1, and with teachers and local community workers in Table 2. Barriers are described for PA, limiting SB and healthy dietary habits, according to parents’ perspectives and teachers’ and local community workers’ perspectives.

### 3.2. Physical Activity

#### 3.2.1. Parents’ Perspectives

Several general barriers to sufficient PA were identified, situated on various levels of the socio-ecological model. These barriers can be found in Appendix A, Section “A. physical activity”. An overview of all barriers to PA can be found in Appendix A, Section A. On the individual level, general barriers were lack of energy/feeling tired, health problems and lack of motivation.


*“I think it is difficult for me to exercise because I end up very tired everyday”*
*(parent—Spain)*

On the interpersonal level, general barriers were lack of time and financial limitations.


*“They (the children) should do more physical activities, but there is no time”*
*(parent—Hungary)*

On the organisational level, lack of public facilities in the neighbourhood (e.g., parks and cycling paths), lack of facilities and space at home, unsafe, old and/or unmaintained public facilities in the neighbourhood and/or uncomfortable or inflexible sports clubs circumstances were general barriers reported by parents.


*“I don’t have a garden … I have a courtyard, but I would like to have a garden”*
*(parent—Belgium)*


*“The park should be safer, there are stones everywhere. A playground (in the park) with stones everywhere”*
*(parent—Belgium)*


*“There are playgrounds close to large boulevards with intense traffic and without green spaces, there is a stadium which however isn’t maintained”*
*(parent—Bulgaria)*


*“Most women complain and say that they do not go (i.e., to the pool) because the cafeteria that is near has a glass wall and people who sit there drinking their coffee are looking at them”*
*(parent—Greece)*

Country-specific barriers to PA were also identified. On the individual level, laziness was mentioned in Finland and Hungary.


*“My son is lazy and slow but so am I”*
*(parent—Finland)*

On the interpersonal level, religion and lack of acculturation was reported in Belgium, no caregiver or supervisor for the children was reported in Finland and Greece and organisational difficulties within the family was reported in Spain and Bulgaria.

On the organisational level, lack of (sports) facilities and space in school was reported in Bulgaria as a barrier, limited opportunity for group activities in sports clubs and a long distance to school was reported in Finland, too crowded public facilities were reported in Spain and fully booked sports clubs were reported in Belgium. Furthermore, closed (public or sports) facilities, lack of advertisement on organised activities, high-threshold sports clubs (e.g., no beginner groups), lack of sports clubs and badly organised (sports) activities were barriers reported in Greece. Finally, an unhealthy school policy was mentioned in Spain and Bulgaria and lack of organised (family) activities for children and/or parents was mentioned in Finland and Greece.


*“And the distance (to work and to school) is so long that I have to go by car”*
*(parent—Finland)*


*“We have a swimming pool but it is always crowded”*
*(parent—Spain)*

On the macro level, a bad economic situation of the country was reported in Greece, while unfavourable weather circumstances were reported in Finland.


*“I could go to the swimming pool in the Olympic Stadium, but to go there it is too far and I have to pay 6 Euro … In Germany, I could pay 3 Euro and swim for as long as I wanted in a very nice pool”*
*(parent—Greece)*

#### 3.2.2. Teachers’ and Local Community Workers’ Perspectives

Although most of the barriers to PA reported by parents were also reported by teachers and/or local community workers, they also provided some additional barriers (see Section “A. physical activity” in Appendix A). On the individual level, additional barriers reported by both teachers and local community workers were lack of motor skills (e.g., not being able to swim or cycle) and lack of knowledge on the importance of sufficient PA. An additional barrier reported by teachers was a misperception regarding transport to school (e.g., showing love by driving the child to school). An additional barrier reported by local community workers was lack of parental skills in encouraging children to be physically active.


*“A lot of them (pupils) do not have a bicycle or cannot ride with it”*
*(teacher—Spain)*


*“I believe that vulnerable families lack knowledge on topics related to physical activity in a great percentage”*
*(local community worker—Greece)*


*“They think that they are doing their child like a favor and they love the child more when they take the child by car to school”*
*(teacher—Finland)*

On the interpersonal level, additional barriers reported by local community workers were the absence of a role model for the children and wanting to render luxury and wealth (i.e., taking the car instead of choosing for active transport).


*“They don’t have a role model. Parents are not doing physical activity”*
*(local community worker—Hungary)*

On the organisational level, unsafe and/or inappropriate playgrounds at school and the absence of active extra-curricular school-based activities were additional barriers reported by teachers.


*“We currently have sufficient space in the school, but the ground is in a bad condition”*
*(teacher—Greece)*

### 3.3. Sedentary Behaviour

#### 3.3.1. Parents’ Perspectives

Several general barriers to limiting SB were mentioned by parents, situated on various levels of the socio-ecological model. These barriers can be found in Appendix A, Section “B. sedentary behaviour”. An overview of all barriers to limiting SB can be found in Appendix A, Section B. On the individual level, these general barriers were the need to relax/rest and unhealthy family traditions (e.g., TV-viewing as family time).


*“At weekdays it’s more like I get quickly to home from school and get children from kindergarten and then I’m also so tired, exhausted and need to rest”*
*(parent—Finland)*


*“I am sitting 8 hours at work … I am also laying in the couch at 00:30–01:00 after children fall asleep … Also, before my children go to bed we watch TV or play board games”*
*(parent—Greece)*

On the organisational level, only one general barrier was identified for limiting SB, i.e., the child(ren) having homework or other educational sitting activities to do (e.g., reading, drawing, etc.).


*“I think that my children spend a lot of time reading and doing homework, not only watching TV”*
*(parent—Spain)*

Unfavourable weather circumstances and the current technology nation in which everyone uses electronic devices (e.g., smartphones and tablets) were general barriers on the macro level.


*“In summer they are outside but in winter they spend a lot of time watching TV”*
*(parent—Spain)*


*“Our nation is becoming completely technological and reducing screen time would be a disadvantage later on, at least I think so”*
*(parent—Bulgaria)*

Country-specific barriers to limiting SB were also identified. On the individual level, health problems were reported as a barrier to limiting SB in Greece, the child preferring sedentary activities (over physical activities) were reported in Belgium and Bulgaria, and being addicted to technology (e.g., smartphone, computer games, TV programs, etc.) was reported in Finland and Greece. 


*“My kids prefer watching TV and being at the computer instead of playing outdoors”*
*(parent—Bulgaria)*

On the interpersonal level, lack of time (and therefore taking the car) was reported as a barrier in Greece. On the organisational level, an inappropriate work environment (e.g., no space for movement breaks or standing desks) was reported in Bulgaria, and a sedentary education system was reported in Bulgaria and Hungary.

#### 3.3.2. Teachers’ and Local Community Workers’ Perspectives

Although most of the barriers to limiting SB reported by parents were also reported by teachers and/or local community workers, teachers and local community workers also provided some additional barriers (see Section “B. limiting sedentary behaviour” in Appendix A). On the individual level, an additional barrier reported by local community workers was lack of knowledge on the importance of limiting SB, and an additional barrier reported by teachers was a misperception regarding SB (i.e., not knowing that prolonged sitting is unhealthy). On the interpersonal level, having bad role models and wanting to render luxury and wealth were additional barriers reported by local community workers. 


*“They do not bicycle for transport because their parents always take the car”*
*(local community worker—Belgium)*

On the organisational level, an additional barrier reported by teachers was an inappropriate school (classroom) environment.


*“We have to take into account that classes are not meant for movement: chairs and tables are heavy, you cannot move them all the time”*
*(teacher—Spain)*

Finally, the negative influence of media and advertisements (e.g., families buying and using technology because advertisements on this are repeatedly presented) was an additional barrier reported by local community workers on the macro level.

### 3.4. Dietary Habits

#### 3.4.1. Parents’ Perspectives

Several general barriers to healthy dietary habits were identified, situated on various levels of the socio-ecological model. These barriers can be found in Appendix A, Section “C. healthy dietary habits”. An overview of all barriers to healthy dietary habits can be found in Appendix A, Section C. On the individual level, these general barriers were unhealthy family traditions (e.g., eating croissants on Sunday and taking dinner late), not liking certain healthy foods (e.g., vegetables and fruits), unhealthy dietary preferences (e.g., preferring sugary soft drinks instead of water) and healthy cooking being demanding and time-consuming.


*“I would like to eat more vegetables … but my children are very picky”*
*(parent—Belgium)*


*“(Changing the eating pattern) for two persons should work, but for a whole family … when you have to cut all those vegetables … I have no time for that”*
*(parent—Belgium)*


*“I don’t like vegetables so I cook it for them (the children) but I don’t eat it myself”*
*(parent—Spain)*

Having bad role models was a general barrier on the interpersonal level.

Country-specific barriers to healthy dietary habits were also identified. On the individual level, lack of energy/feeling tired, unhealthy eating out of boredom and health problems were barriers reported in Greece.


*“I am too tired to prepare the fish and cook it”*
*(parent—Greece)*

Lack of parental skills in monitoring the eating behaviours of the child was mentioned in Greece and Hungary, and lack of parental self-efficacy in monitoring healthy dietary habits was reported in Belgium and Greece.

On the interpersonal level, having unhealthy cultural habits was mentioned in Belgium as a barrier, unhealthy eating and drinking being a reward was mentioned in Greece, and the availability of unhealthy foods/drinks was reported in Greece and Hungary.


*“I am always under pressure and overstressed, especially at winter without any breaks, and this makes me to reward myself with unhealthy food”*
*(parent—Greece)*

On the organisational level, a long distance to healthy food facilities (e.g., organic food shops) was reported in Greece, and an unhealthy school policy (e.g., school caterer that provides unhealthy and low quality food, no rules about which snacks are allowed, etc.) was reported in Bulgaria and Greece.


*“The snack that is given to the children in up to fourth grade may include fruits and vegetables, but only if you see those vegetables … Well, it doesn’t look nice, it has no taste”*
*(parent—Bulgaria)*

On the macro level, the negative influence of media and advertisements was reported in Finland, and seasonal factors (i.e., certain fruits and vegetables are more prevalent in a specific season) and the bad economic situation of the country were reported in Greece. Lastly, the high cost of healthy foods compared to fast food was reported as a country-specific barrier in Greece and Spain.


*“And this (the idea that weekends are for unhealthy treats) is kind of kept up by advertisements as for example this ‘Pizza Friday thing’”*
*(parent—Finland)*


*“I would like to eat more kind of foods, I think that we always eat the same but is because of the economy”*
*(parent—Spain)*

#### 3.4.2. Teachers’ and Local Community Workers’ Perspectives

Some additional barriers were identified for healthy dietary habits (see Section “C. healthy dietary habits” in Appendix A). On the individual level, lack of cooking skills was an additional barrier reported by local community workers, and lack of knowledge on the importance of healthy dietary habits was an additional barrier reported by both teachers and local community workers.


*“I think that they (parents) often don’t know what is healthy … and that is actually the problem”*
*(local community worker—Belgium)*

On the interpersonal level, wanting to render luxury and wealth (e.g., drinking sugary soft drinks or energy drinks from a popular brand) was an additional barrier reported by local community workers.


*“It is also something like social status. For example, it is cooler to drink cola or an energy drink from a popular brand”*
*(local community worker—Belgium)*


*“The cola is unhealthy, but for them it means a status symbol”*
*(local community worker—Hungary)*

A low school budget to organise healthy eating activities at school was an additional barrier reported by teachers on the organisational level.


*“The price-quality relation (of the school lunch) makes you think ‘if this was a work place cafeteria, many would choose another place’”*
*(teacher—Finland)*

## 4. Discussion

The present study investigated the barriers experienced by young families (parents and children) from vulnerable areas towards PA, limiting SB and healthy dietary habits, which are three important lifestyle behaviours in the prevention of T2D [4,5,6]. These barriers were investigated from multiple perspectives. For this, focus groups were conducted across six European countries with parents of young (primary school age) children living vulnerable areas, and with primary school teachers and local community workers who often come into contact with these families.

The first aim of the study was to investigate these barriers according to the perspectives of parents, and to explore whether these barriers are general (i.e., reported in four countries or more) or country-specific (i.e., reported in only one or two countries). This study identified general barriers which were situated on all levels of the socio-ecological model of health behaviour: the individual level (e.g., lack of energy and motivation for PA, unhealthy family traditions for limiting SB and unhealthy dietary preferences for healthy dietary habits); interpersonal level (e.g., lack of time for PA and bad role models for dietary habits); organisational level (e.g., lack of facilities in the neighbourhood and at home for PA and having homework for limiting SB) and macro/public policy level (e.g., the current technology nation and unfavourable weather circumstances for limiting SB). Some of these general barriers (e.g., lack of energy, time and facilities for PA, unhealthy family traditions for limiting SB and healthy cooking being time-consuming for healthy dietary habits) were also reported in previous, nationally conducted studies investigating barriers experienced by socially disadvantaged families [20,21,22,23,24,25], which might imply that these barriers indeed play an important role in the lifestyle behaviours of families living in vulnerable areas. Additionally, this study shows that these barriers apply at a European level, as these barriers were reported across different European countries. Furthermore, the barriers found in previous studies were mostly individual and interpersonal barriers, while the present study also found barriers on other levels of the socio-ecological model (e.g., organisational and macro/public policy level). This is probably because this study investigated barriers from a broad, multilevel perspective using a socio-ecological framework, not only investigating individual characteristics and proximal social influences but also the broader community, organisational and policy influences.

That barriers were identified on all levels of the socio-ecological model, might imply that all these levels influence the health behaviours of young families living in vulnerable areas, and need to be taken into account when developing an intervention for this target group. Therefore, we suggest for future lifestyle interventions aiming to promote a healthy lifestyle in young families from vulnerable areas to include multiple intervention components, covering the various levels of the socio-ecological model: one component focusing on the family, one component focusing on the school (environment), and one component focusing on the broader community. This is also in line with previous studies showing that multicomponent interventions (i.e., including multiple levels) are generally associated with better outcomes [31,43,44]. The family-component could, for example, focus on increasing families’ motivation, knowledge and skills (i.e., cooking skills, parental skills, motor skills). Furthermore, there could be focused on changing unhealthy family traditions and preferences, and on time-management (e.g., how to implement a healthy lifestyle in daily life). Finally, parents could be encouraged to be a good role model for their children. The school-component could focus on promoting movement breaks, providing less educational sedentary tasks and/or homework and on improving the school (class and playground) environment by making it safer and more suitable for children. The community-component could focus on informing families about the available safe and maintained public facilities (e.g., parks). Furthermore, sports clubs and/or the community could be encouraged to provide activities that meet the needs of this target group (e.g., providing swimming hours for women only).

Besides general barriers, the present study also identified country-specific barriers. We therefore recommend, when aiming to prevent T2D in young European families in vulnerable areas, to develop (and implement) a cross-European lifestyle intervention including similar components, design and methodology, which also has room for local, country-specific adaptations. This is in line with other studies also making this recommendation for intervention development in the general population [45,46]. In this way, a potentially cost-effective intervention is developed in an efficient way, with the possibility to compare the intervention effects across European countries. However, a larger sample, preferably with focus groups from different vulnerable areas, is needed to formulate meaningful country-specific conclusions and recommendations.

A second aim of the study was to investigate if teachers and local community workers provide extra information in addition to the barriers reported by parents. Most of the barriers reported by parents were also reported by teachers and/or local community workers. However, teachers and local community workers also provided barriers that (they believe) are experienced by families living in vulnerable areas, which were not reported by the parents. Additional barriers reported by local community workers were mainly situated on the individual level (e.g., lack of knowledge on healthy behaviours, lack of parental skills in monitoring PA, and lack of cooking skills) and interpersonal level (e.g., bad role models and wanting to render luxury and wealth), additional barriers reported by teachers were mainly situated on the individual (e.g., misperceptions towards PA and limiting SB) and organisational level (e.g., unsafe and/or inappropriate school environment, low school budget for healthy activities, and the absence of extra-curricular school-based activities). This shows the additional value of combining the perspective of parents with the perspective of other important stakeholders such as teachers and local community workers, and this information can be used for future intervention development in this target group. However, as only one focus group with teachers and one focus group with local community workers was conducted in each country, more research is needed investigating the perspectives of teachers and local community workers across European countries to draw meaningful conclusions on this.

A limitation of this study is that some of the countries did not reach the conventional group size of minimum six participants per session, and that not all focus groups were gender-mixed. More specifically, the majority of focus group participants were mothers (i.e., 86%). Although mothers are generally considered to be the primary caregivers with the most insight into the health behaviours of their family and children, future research could include (more) fathers and study also their perception on the barriers towards health behaviours experienced by their families. A second limitation is that in each country only one focus group with teachers and one focus group with local community workers was conducted, which means that saturation was probably not achieved in these focus groups. Therefore, it was not possible to compare the reported barriers of teachers and local community workers between the different countries, and to formulate country-specific conclusions. Third, parents and teachers were recruited from schools located in vulnerable areas. However, it may be the case that children attend a school in a vulnerable area but come from an area which is not vulnerable. A fourth limitation is that only a limited number of barriers have been identified for limiting SB. This may be explained by the fact that the focus group participants possibly confused the term “sedentary behaviour” with the term “physical inactivity”, even though these terms were clearly explained at the beginning and during the focus group discussion. It is however important for participants to clearly understand this difference to be able to report barriers for these distinct behaviours. Consequently, future research conducting focus groups is recommended to provide education on (the difference between) these terms at the beginning of a focus group discussion. Furthermore, we also identified only a limited number of work-related barriers. Future studies can address this issue by, for example, including a work-related topic in the focus group interviews, which was not done in the present study. Finally, the semi-structured interview guide was a relatively long guide, but the focus groups lasted at most 90 min. This study also has some strengths. First, focus group interviews were conducted in six European countries, which makes it possible to detect international as well as country-specific barriers. Another strength is that this study included multiple perspectives on the barriers experienced by young families living in vulnerable areas (i.e., the perspective of the parents, teachers and local community workers). To date, no study has been conducted which has investigated these perspectives in one study. Furthermore, a standard protocol and a semi-structured interview guide was developed and used for conducting the focus groups, to ensure standardisation across the six European countries.

## 5. Conclusions

The present study identified barriers towards three important health behaviours related to T2D, experienced by young families living in low socio-economic areas (SES), which were situated on various levels of the socio-ecological model of health behaviour. Therefore, it is recommended for future interventions to include multiple components (e.g., focusing on the family, school and community), covering all levels of the socio-ecological model. Several of these barriers were generally reported across the six participating European countries, but also some country-specific barriers were identified. Future interventions targeting young families living in vulnerable areas could develop a standard European intervention that has room for country-specific adaptations. Furthermore, teachers and local community workers reported some barriers that were not reported by parents. This shows the additional value of combining parents’ perspectives with the perspectives of other stakeholders’ perceptions (e.g., teachers and local community workers), and the importance of including them when developing a lifestyle intervention targeting young families living in vulnerable areas. However, more research is needed to investigate the perspectives of teachers and local community workers across European countries to identify similarities across countries as well as country-specific differences.

## Figures and Tables

**Table 1 ijerph-15-02840-t001:** Descriptive information of the focus groups with parents across the six European countries.

Country	Vulnerable Area	Low Educated Participants ^a^ (%)	Range of Participants across the Focus Groups	Total Participants	Sex (% Female)
Belgium	Lokeren	44	4–7	18	78
Bulgaria	Varna	25	6–8	20	90
Finland	Askisto (Vantaa) and Ruskeasuo (Helsinki)	15	6–7	20	85
Greece	Kallithea (Athens)	40	4–6	15	93
Hungary	Nyirmihalydi and Debrecen	23	8–11	28	82
Spain	Zaragoza	73	4–5	14	86
		Mean: 37	Min. 4, Max. 14	Total: 115	Mean: 86

Note: ^a^ ≤14 years of education.

**Table 2 ijerph-15-02840-t002:** Descriptive information of the focus group with teachers and local community workers within the six European countries.

Country	Vulnerable Area	Mean Years of Experience with Target Group	Sex (% Female)	Total Participants
		**Teachers**		
Belgium	Ghent	3.4	100	7
Bulgaria	Varna	26.7	100	10
Finland	Kirkonkylä (Nurmijärvi)	15.7	100	6
Greece	Kallithea (Athens)	21.3	75	8
Hungary	Debrecen	27.8	100	6
Spain	Zaragoza	6.7	75	8
		*Mean: 16.9*	*Mean: 92*	*Total: 45*
		**Local community workers**		
Belgium	Ghent	7.1	71	7
Bulgaria	Sofia	28	75	8
Finland	Tikkurila (Vantaa)	6.4	80	5
Greece	Kallithea (Athens)	9.6	86	7
Hungary	Nyírmihálydi	16.4	88	8
Spain	Zaragoza	3.9	100	6
		*Mean: 13.5*	*Mean: 83*	*Total: 41*

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
