# Peer review of "Barriers from Multiple Perspectives Towards Physical Activity, Sedentary Behaviour, Physical Activity and Dietary Habits When Living in Low Socio-Economic Areas in Europe. The Feel4Diabetes Study"

_ijerph, 2018, doi:10.3390/ijerph15122840_

Round 1

Reviewer 1 Report

Please find reviewer's comments attached.

Author Response

We thank the reviewer for his/her appreciation of our work and valuable comments. We have addressed the comments in attachment. Furthermore, the manuscript has been passed to a native English speaker for additional editing. 

Reviewer 2 Report

Reviewer comments - ijerph-379416

Overall comments:

The manuscript is generally well written and structured and the topic is interesting. The analysis stays on a descriptive (manifest) level, which is ok if that was the authors’ intention. However, it is necessary that the authors add information and references to describe how they have ensured the quality and trustworthiness of the analysis and the results.

Major concerns:

The qualitative process is not sufficiently described and the authors should add some references in the methods section to clarify and justify their choices in data collection and analysis and also thoroughly discuss them in the discussion. I suggest that the authors follow the recommendations in Elo et al 2014, Qualitative Content Analysis: A Focus on Trustworthiness.

I am not convinced that the study design can give sufficient information to conclude “country-specific” barriers based on the results at the individual level. I believe that they are not comprehensive enough based on the study sample. As I have understood it, the authors’ choice is a deductive approach with a manifest analysis, i.e. to describe what the informants actually say and stay very close to the text. (In contrast to latent analysis, which extends to an interpretive level and an underlying meaning?) This means that to be able to come to conclusions about the specific countries from the individual level a much larger sample would be needed, with focus groups from different SES areas etc.
For example, on line 193: Do you really mean that being lazy is something specific for Finland and Hungary? Or having organizational difficulties in the family is specific for Spain and Bulgaria?

Line 145: I suggest that the authors provide additional information about the researchers, such as profession and previous experience, to clarify their pre-understanding. 

Lines 198 and 281: Why are quotes about dietary habits are used as examples for PA and SB?

The use if deductive content analysis should be stated already in the abstract.

Minor concerns:

The title needs revision: Physical activity occurs twice and the authors have to revise the expression “barriers towards sedentary behavior”. Maybe change to the expression used in line 70: “barriers towards limiting sedentary behavior”.

Line 82: New abbreviation: LCW

Line 95: Qualitative research aims to investigate experiences, perceptions attitudes etc and not just to list something - the authors should therefore consider revising the aim and exclude “which”. 

Line 123: (i.e. 3 focus groups… This is unnecessary information. It was described in the previous paragraph.

Line 126: The authors should use “dietary” habits consistently.

Line 127: I can’t find the Interview guide appendix.

Line 172: Table 1. A definition of low education should be added. The numbers for participants in Bulgaria, Greece and Spain do not match for 3 groups in each country.

Author Response

(The authors gave the same response as above.)

Reviewer 3 Report

General comments

Thank you for your submission which ahs provided me with the opportunity to review the manuscript.  I commend the researchers for collaborating to produce this multi-cite, cross-country study - I appreciate how challenging this can be.  I have made a variety of observations below which I do hope you will be able to make sense of and which I hope you find useful in helping you to produce a stronger manuscript.

Generally, I felt the manuscript is well written using standard English, however, there are occasions when syntax needs attention.  Similarly, some sentences require effective use of punctuation to aid understanding.  Many of these observations are detailed in the respective manuscript sections below.  It would be prudent to pass the manuscript to a native English speaker for additional editing.

Introduction

Lines 73 and 75 – what is meant by a ‘good view’?  I suggest this is rephrased and suspect that the authors are implying that teachers are well placed to comment upon potential barriers

It is unclear from the introduction why young families (children attending primary school) are the focus.  A stronger argument is needed here.   Why not families of secondary school aged children?  Is the younger age range of particular importance in terms of T2DM prevention?

It is clear that the research has been very much framed by a social-ecological model.  However, the introduction does not provide any commentary on SEM to provide a foundation, and rationale, for the adoption of SEM in the analysis, which is needed.

Methods

Participants

How many participants were recruited of each category (i.e., parents, teachers and community workers)? They are presented according to country in the results section but there is a need for totals to be presented as part of the methods section.

Can you please confirm whether individual participant schools, from which parents and teachers were recruited, were selected on the basis that the majority of children enrolled at the school are from low SES households or were the schools in areas of social and economic deprivation?  It may be that children attend a school in a deprived area when they and their parents may live in an area and/or household which is not low SES.  This may need to be acknowledged as a limitation in the discussion.

Procedure

How ‘large’ were focus groups?  How many participants per focus group?  Again, they are presented according to country and participant category in the results section but there is a need for the range overall to be presented as part of the methods.

Line 136 - How were educational level established? 

Results

Line 194 – ‘lack of acculturation and religion’.  Can this be clarified and rewritten in the manuscript as necessary.  What is being implied - a lack of acculturation and a lack of religion OR religion and lack of acculturation?

This comment is for consideration.  You enable easier comparison between the perspectives of parents and those of teachers and community workers, would it be advantageous to organise the results section according to behaviour (PA, SB and dietary) then the levels of socio-ecological model, in which example focus group excerpts from both parents, and teachers and community workers are presented together?  This is as opposed to the current structure (i.e., behaviours; participant groups; SEM levels)

Discussion

Line 350-353 – this sentence is a little confusing.  It begins mentioning PA, then outlines barriers for SB, before going back to barriers to PA.

Lines 346-353 – it is unclear why there are citations in these sentences, as there is no explicit reference to comparing and contrasting the findings of the current study with previous literature.

Lines 401-404 – this sentence needs punctuation to aid the reader’s comprehension.

Line 428 – is the bigger concern here that there was a very small number of male participants across the participant groups, as opposed to the lack of males from the focus groups? And, what does this mean for the application of the current findings and the development of future interventions? Or are there arguments to present that females are still the primary care givers?  Please consider this.

Line 442 – ‘Up to now, no study has been executed yet investigating these multiple perspectives in one study’ – this sentence needs to be rewritten (would ‘To date, no study has been conducted which has investigated these multiple perspectives in one study’ communicate the point more clearly?)

Author Response

(The authors gave the same response as above.)

Round 2

Reviewer 1 Report

Review of manuscript ijerph-379416

“Barriers from multiple perspectives towards physical activity, sedentary behaviour, physical activity and dietary habits when living in low socio-economic areas in Europe. The Feel4Diabetes study.”

Meanwhile, I have read the revised manuscript and it has much improved. The authors have paid close attention to the issues mentioned, and have handled them mostly adequately.

I still would have preferred a more thorough theoretical description of the socio-ecological perspective in the introduction, being it a theoretical framework. What follows, is that the discussion could benefit from more closely linking the results to the socio-ecological perspective, thus in fact utilize the model as a theoretical framework. In my opinion, the socio-ecological model appears applied to provide structure to the analysis, and not as a theoretical basis contributing in the interpretation of the results.

Due to the number of analytic levels, I also think the results could have been better arranged in order to make them easier to read. The quotes following the text are not necessarily on the same analytic level as the text before, which would likely confuse the reader. Perhaps a figure could be of help. However, this should not prevent the paper from being published.

The revised version of the paper has improved considerably. Pending the authors’ handling of the issue of theoretical basis, the manuscript would seem ready for publication.

Author Response

We thank the reviewer for his/her appreciation of our work and valuable comments, and for considering the manuscript for publication. We have addressed the reviewer's (minor) comments and changes made in the manuscript are highlighted in yellow. To answer the reviewer's first final comment: we indeed used the socio-ecological model to provide structure to the analysis/results, and not as a theoretical basis/framework. Therefore, we did not add a more thorough theoretical description of the socio-ecological model in the introduction and did not use the model for interpretation of the results in the discussion. Instead, we revised the parts mentioning the use of the socio-ecological model as a theoretical framework, and described its use more correctly/appropriately (i.e. as a guide/perspective for investigating the barriers and interpretation of results). Last, we have arranged the quotes better so that they are following the same analytical level as the text before. Quotes that have been replaced are highlighted in yellow throughout the results section.

Introduction, line 104-105.

The present study will be guided by the socio-ecological model of health behaviour to investigate these barriers. 

Introduction, line 109-112.

The first aim is to investigate, from a socio-ecological perspective, the barriers experienced by young families living in vulnerable areas towards healthy lifestyle behaviours according to the perspective of parents, and to explore whether these barriers are general or country-specific.

Discussion, line 383-385.

The present study investigated the barriers experienced by young families (parents and children) from vulnerable areas towards PA, limiting SB and healthy dietary habits, which are three important lifestyle behaviours in the prevention of T2D [4-6]

*In this sentence we omitted the use of a socio-ecological model as a theoretical framework*

Reviewer 2 Report

I am satisfied with the authors' responses and changes and have only two minor further comments. Page 4 line 153. Sentence starting with A protocol... should be checked for grammar. Page 4 line 181. I would like the authors to also include information about the doctoral researchers education/profession so that the reader gets better information about the pre-understanding of the two persons who conducted the analysis.

Author Response

We thank the reviewer for his/her appreciation of our work and valuable comments, and for considering our manuscript for publication. We have addressed the reviewer's last comments (i.e. checking/revising sentence for grammar and adding information on researchers’ education/profession). Changes made in the manuscript are highlighted in yellow.

Methods, procedure, line 153-155

A protocol with three semi-structured interview guides (one for parents, one for teachers and one for community workers) was developed to ensure standardization across the six European countries

Methods, data analyses, line 181-183

[…] barriers were coded by two independent doctoral researchers (VVS; Master in Movement and Sports Sciences, Ghent University, Belgium; and JL; Master in Experimental and Theoretical Psychology, Ghent University, Belgium; ICC: 0.86)